# Membrane Association and Topology of Citrus Leprosis Virus C2 Movement and Capsid Proteins

**DOI:** 10.3390/microorganisms9020418

**Published:** 2021-02-17

**Authors:** Mikhail Oliveira Leastro, Juliana Freitas-Astúa, Elliot Watanabe Kitajima, Vicente Pallás, Jesús Á. Sánchez-Navarro

**Affiliations:** 1Unidade Laboratorial de Referência em Biologia Molecular Aplicada, Instituto Biológico, São Paulo, SP 04014-002, Brazil; juliana.astua@embrapa.br; 2Instituto de Biología Molecular y Celular de Plantas, Universidad Politécnica de Valencia-Consejo Superior de Investigaciones Científicas (CSIC), 46022 Valencia, Spain; vpallas@ibmcp.upv.es; 3Laboratório de Fitopatologia, Embrapa Mandioca e Fruticultura, Cruz das Almas, BA 44380-000, Brazil; 4Departamento de Fitopatologia e Nematologia, Escola Superior de Agricultura Luiz de Queiroz, Universidade de São Paulo, Piracicaba, SP 13418-900, Brazil; ewkitaji@usp.br

**Keywords:** membrane association, topology, cilevirus, movement protein, p29 capsid protein

## Abstract

Although citrus leprosis disease has been known for more than a hundred years, one of its causal agents, citrus leprosis virus C2 (CiLV-C2), is poorly characterized. This study described the association of CiLV-C2 movement protein (MP) and capsid protein (p29) with biological membranes. Our findings obtained by computer predictions, chemical treatments after membrane fractionation, and biomolecular fluorescence complementation assays revealed that p29 is peripherally associated, while the MP is integrally bound to the cell membranes. Topological analyses revealed that both the p29 and MP expose their N- and C-termini to the cell cytoplasmic compartment. The implications of these results in the intracellular movement of the virus were discussed.

## 1. Introduction

Citrus leprosis disease, caused by viruses belonging to the genera *Cilevirus* and *Dichorhavirus*, has economic importance in the Americas, especially in citrus groves in Brazil. The infection is characterized by the formation of chlorotic and necrotic circular localized lesions in citrus leaves, fruits, and stems [1].

Citrus leprosis virus C (CiLV-C) is a member of genus *Cilevirus,* family *Kitaviridae,* and a prevalent citrus leprosis-associated virus in South and Central America [1], with the exception of Colombia. Although CiLV-C was reported in that country more than 10 years ago, this virus is currently rarely found in Colombian citrus orchards, where it was replaced by citrus leprosis virus C2 (CiLV-C2) [2,3,4,5]. CiLV-C and CiLV-C2 are the only accepted members of genus *Cilevirus*.

Cileviruses have a bisegmented positive ssRNA genome carrying a 5′-cap structure and a 3′-poly (A) tail. For CiLV-C, its RNA1 codes for the protein precursor of the RNA-dependent RNA polymerase (RdRp) and the capsid protein (p29) [6,7,8], while the RNA 2 codes for an RNA silencing suppressor (RSS) protein (p15) [9], the viral movement protein (p32) [10], and the putative glyco (p61) and matrix (p24) proteins [7,11]. In addition to p15 RSS activity, the p29 and p61 are proteins that also show the ability to suppress RNA silencing [9]. CiLV-C2 RNA1 and RNA 2 present a nucleotide identities of 58 and 50%, respectively, compared to the corresponding CiLV-C RNA sequences. Additionally, CiLV-C2 RNAs have a longer 3′ UTR and an extra ORF (p7) in RNA2 [2].

Membrane-associated viral proteins can induce substantial cellular remodeling in the processes of viral replication and virion assembly. This intracellular disorder is usually associated with changes in organelles to form viral replication sites [12,13]. Viral proteins can also interact with cell membranes to facilitate the intracellular viral spread [14,15,16]. Association between viral movement components and host membranes seems to be an essential factor for virus transport, being a feature constantly identified in this class of viral proteins [7,17,18,19,20,21]. Despite the fact that recent studies have clarified some functional aspects for the CiLV-C proteins, no similar information is available for other accepted or tentative cileviruses, as passion fruit green spot virus (PfGSV). This group of viruses has been rather poorly investigated and thus, generating new data about other species of agronomic importance, such as CiLV-C2, can represent an important advance in understanding the citrus leprosis pathosystem.

Therefore, in the present study, we examined biochemical properties of the capsid (p29) and movement (MP) proteins encoded by CiLV-C2. Here, we reported the association of CiLV-C2 MP and p29 with biological membranes. Our findings obtained by computer predictions, chemical treatments after membrane fractionation, and biomolecular fluorescence complementation assay revealed that the p29 is peripherally associated, while the MP is integrally bound to the cell membranes. Topological analyses revealed that p29 and MP expose their N- and C-termini to the cell cytoplasmic compartment. These results allowed us to propose a topological model of the association of CiLV-C2 p29 and MP with cell membranes.

## 2. Materials and Methods

### 2.1. DNA Manipulation

For the membrane association assay, the *p29* and *MP* genes [10] were amplified with specific primers carrying the *Nco*I/*Nhe*I sites using the TaKaRa Taq DNA polymerase (TaKaRa Bio Inc) following the manufacturer’s specifications. All amplified genes were fused at their C-terminus with the human influenza hemagglutinin (HA) epitope. To do this, the genes were cloned into the vector pSK35-TSWVNSm:HA-PoPit [18], replacing the *NSm* gene. The correct in-frame insertions were confirmed by plasmid DNA sequencing. The expression cassettes corresponded to each CiLV-C2 isolated gene flanked by the 35S constitutive promoter from the cauliflower mosaic virus (CaMV) and the terminator from the potato proteinase inhibitor (PoPit) [18]. Next, the expression cassettes were cloned into the pMOG800 binary vector by using the *Hind*III restriction site.

For bimolecular fluorescence complementation (BiFC) assay, the *p29* and *MP* genes were amplified with specific primers, containing the *Nco*I/*Nhe*I restriction sites, and cloned into the pSK BiFC plasmids [18,22], which permitted the fusion of the N- or C-yellow fluorescent protein (YFP) fragments at their N- or C-termini. The resultant expression cassettes were subcloned into the pMOG800 binary vector as previously reported [7].

The pMOG vectors carrying the chrysanthemum stem necrosis orthotospovirus (CSNV) NSm, tobacco mosaic virus (TMV) 30K MP, and leader peptidase (Lep) proteins with the HA epitope fused at their C-terminus and the unfused green fluorescent protein (GFP) were described in previous works [7,18].

### 2.2. Computer Analysis

Computer analysis from deduced amino acid sequences of the CiLV-C2 p29 (YP_009508071.1) and MP (YP_009508075.1), CiLV-C MP (ABC75825.1), CSNV NSm (AII20574.1), TMV 30K MP (AAD19281.1), and also for the Lep (WP_112485673.1) and GFP, were performed using computer tools MPEx [23], TMHMM server version 2.0 (http://www.cbs.dtu.dk/services/TMHMM/), ΔG prediction server version 1.0 (http://dgpred.cbr.su.se/index.php?p=home), HMMTOP version 2.0 (http://www.enzim.hu/hmmtop/index.php), OCTOPUS (https://octopus.cbr.su.se), SOUSI (http://harrier.nagahama-i-bio.ac.jp/sosui/), and TMpred (https://embnet.vital-it.ch/software/TMPRED_form.html).

### 2.3. Subcellular Fractionation, Chemical Treatment, and Western Blot Analyses

*Nicotiana benthamiana* leaves were agroinfiltrated (OD_600_ = 0.5) with the pMOG800 construct carrying the CiLV-C2 *p29* or *MP* ORFs fused to the HA epitope. As controls, we used *N. benthamiana* leaves agroinfiltrated with constructs expressing the HA-tagged Lep protein, unfused GFP, and HA-tagged NSm of CSNV. At 3 days post-infiltration (dpi), approximately 1.5 g of *N. benthamiana* leaves expressing the tested proteins were ground in lysis buffer (20 mM de HEPES, pH 6.8; 150 mM potassium acetate; 250 mM mannitol; 1 mM MgCl_2_, and 50 µL of protease inhibitor cocktail for plant cell, Sigma–Aldrich, St. Louis, MO, USA). The homogenate was clarified by low centrifugation at 3000× *g* for 10 min at 4 °C, then the obtained supernatant was ultracentrifuged at 40,000× *g* for 40 min at 4 °C to yield the soluble (S30) and the crude (P30) microsomal fraction as reported previously [7,24]. Next, the P30 membrane-rich fraction was resuspended in lysis buffer, divided in six equal parts, and subjected to the chemical treatments with lysis buffer, 100 mM Na_2_CO_3_ (pH 11), 2, 4, or 8 M urea, and held for 30 min on ice. All fractions were analyzed by western blot in 12% SDS-PAGE gels as referred to previously [18] using an anti-HA antibody or an anti-GFP antibody (Thermo Fischer Scientific, Waltham, MA, USA).

For Triton X-114 analysis, the P30 fraction was resuspended in the lysis buffer containing 1% Triton X-114 and incubated on ice for 30 min. The mixture was clarified by centrifugation at 10,000× *g* for 20 min at 4 °C. The resultant supernatant was incubated at 37 °C for 10 min and centrifuged at 10,000× *g* for 10 min at room temperature to form the aqueous (AP) and organic phases (OP). Finally, the OP was resuspended in lysis buffer with the same volume obtained in the aqueous phase, and the fractions were analyzed by western blot in 12% SDS-PAGE gels, as above mentioned. Percentage values of relative protein accumulation was measured using Fiji ImageJ program Ver 2.0 with ISAC plugin. A reference value equivalent to 100% was given based on the pixel quantification of the controls treated with lysis buffer.

### 2.4. Bimolecular Fluorescence Complementation Assays

*Agrobacterium tumefaciens* cultures (C58) transformed with the constructs carrying the p29 or MP with the N-YFP or C-YFP fragments were co-infiltrated (OD_600_ = 0.4) with agrobacterium cultures carrying BiFC constructs containing the counterparts of the YFP addressed to cytosol/nucleus (N-YFPcyt and C-YFPcyt) or to the lumen of the endoplasmic reticulum (ER) (C-YFPer and N-YFPer) in *N. benthamiana* leaves. The plants were maintained in a FITOTRON growth chamber under conditions of 24 °C day/18 °C night at 16 h light/8 h dark and 70% humidity regime. The preparation for agroinfiltration was conducted as described previously [7]. At 4 dpi, fluorescence reconstitution was observed. To increase the expression, in order to allow a better visualization of the fluorescence signal, all protein pair combinations were co-expressed with the silencing suppressor HCPro from the tobacco etch virus.

### 2.5. Confocal Laser Scanning Microscopy

Fluorescence images of the leaf discs from *N. benthamiana* were captured with the aid of a confocal laser scanning microscope Zeiss LSM 780 model. YFP was excited at 514 nm and emission was captured at 520–560 nm. The images were prepared using Fiji ImageJ program (version 2.0r).

## 3. Results and Discussion

### 3.1. The CiLV-C2 p29 and MP Are Membrane-Associated Proteins

Computer analysis from deduced amino acid sequences of the CiLV-C2 p29 and MP was performed using several computer tools. In the analysis, we also included the deduced amino acid sequences of the CSNV NSm and TMV 30K MP, which have been shown to be peripheral membrane proteins [18,19], as well as for the Lep, CiLV-C MP and GFP, two integral membrane-, and one non-membrane-associated proteins, respectively [7,19,24]. Highlighted, they revealed the presence of three hydrophobic regions (HR) for CiLV-C2 MP that encompass the residues 79–97 (HR1), 113–131 (HR2), and 167–185 (HR3), and two for p29: 3–21 (HR1) and 178–196 (HR2) (Figure 1 and Appendix A). Two transmembrane-spanning domains were predicted for the CiLV-C2 MP: 76–98 (TM1) and 160–170 (TM2), which overlap in part with the predicted HR1 and HR3, respectively (Appendix A and Appendix A). Complete predictions for all proteins are presented in Appendix A. Taken together, these analyses suggested that CiLV-C2 p29 and MP are potentially membrane-associated proteins.

In order to confirm these predictions and examine the membrane association of CiLV-C2 p29 and MP, we prepared subcellular microsomal fraction from *N. benthamiana* leaves transiently expressing the CiLV-C2 p29 or MP ORFs fused to the HA epitope. As controls, we used *N. benthamiana* leaves agroinfiltrated with constructs expressing the HA-tagged Lep protein, unfused GFP, and HA-tagged NSm of CSNV, which corresponded to the transmembrane-, cytosolic-, and membrane-associated proteins, respectively. High-speed ultracentrifugation was performed to separate the plant leaf lysed extract, containing the abovementioned proteins, into pellet (P30) and supernatant (S30) fractions. Treatment with Na_2_CO_3_ is known to render microsomes into membrane sheets, releasing soluble luminal proteins [20], while urea treatment should release all polypeptides bound to the membrane, except for the integral membrane proteins [21,24]. We observed that the HA-tagged p29 and MP remained mostly associated with the P30 membranous fraction after Na_2_CO_3_ treatment (Figure 2A, P30 78% for p29, and P30 99% for MP), suggesting that these proteins are tightly associated with the membrane. The Lep and NSm controls remained in the membranous fraction (Figure 2A, P30 100% Na_2_CO_3_) and the majority of the GFP protein remained in the soluble fraction (Figure 2A, P30 79% Na_2_CO_3_), as expected. After the urea treatments and specially after the more aggressive treatment with 8 M urea, the HA-tagged MP and Lep proteins remained in the pellet fraction (100% for both proteins), indicating that the CiLV-C2 MP is fully integrated to cell membranes. On the other hand, a considerable proportion of the HA-tagged p29 protein was observed to be associated with the soluble fraction (43%) after 8 M urea treatment. Similar behavior was observed for the CSNV NSm control (79%, 8 M) suggesting that p29 is peripherally associated with the membrane.

To confirm the cell membrane association of the p29 and MP proteins, the P30 membrane-rich fraction was treated with Triton X-114. This treatment generated two phases, the aqueous (AP) and organic phases (OP), in which the integral membrane proteins should be portioned into the OP, meanwhile the AP should contain non-integral membrane proteins and soluble proteins [25]. As expected for an integral membrane protein, the Lep and MP were recovered from the OP (Figure 2B); meanwhile, p29 and GFP were mostly recovered from the AP (77% for p29 and 95% for GFP). Taken together, these findings indicate that the MP behaves as a membrane-spanning protein and the p29 is physically—but not integrally—associated with membranes.

### 3.2. The CiLV-C2 p29 and MP Have the N- and C-Termini Exposed to the Cell Cytoplasmic Compartment

In the next step, the subcellular compartments in which the N- or C-termini of the p29 and MP proteins are exposed were analyzed by BiFC. *A. tumefaciens* cultures (C58) transformed with the constructs carrying the p29 or MP with the N-YFP or C-YFP fragments were co-infiltrated with agrobacterium cultures carrying BiFC constructs containing the counterparts of the YFP addressed to cytosol/nucleus (N-YFPcyt and C-YFPcyt) or to the lumen of the endoplasmic reticulum (C-YFPer and N-YFPer) in *N. benthamiana* leaves. All protein pair combinations are shown in Appendix A. At 4 dpi, the reconstitution of the fluorescent-competent YFP structure was visualized. Fluorescence reconstitution was visualized when the two YFP halves were co-expressed in the endoplasmic reticulum (ER) or cytosol (positive controls: N-YFPcyt + C-YFPcyt or N-YFPer + C-YFPer; Figure 3i,ii); however, no fluorescence signals were observed when the two YFP halves were co-expressed in different subcellular compartments (negative controls: N-YFPcyt + C-YFPer or N-YFPer + C-YFPcyt; Figure 3iii,iv). When p29, carrying the N-YFP fused at its N- (N-YFP-p29) or C-terminus (p29-N-YFP), was co-expressed with the C-YFPcyt, the reconstitution of the YFP fluorescence was observed in aggregates through the cytoplasm (Figure 3v,vi). Cytoplasmic YFP fluorescence was observed for the MP, carrying the N-YFP fragment fused at its N- (N-YFP-MP) or C-terminus (MP-N-YFP-MP), when co-expressed with C-YFPcyt in *N. benthamiana* leaves (Figure 3ix,x). The fluorescent signal was also visualized into the nucleus for the MP-N-YFP + C-YFPcyt combination (Figure 3ix, red arrows). No fluorescence signal was visualized when the p29 and MP constructs were co-expressed with the N-YFP or C-YFP fragments targeted to the ER (Figure 3vii,xi). Taken together, these findings indicate that both the N- and C-termini of p29 and MP are exposed to the cytoplasmic face. Furthermore, when MP is present within the nucleus, its C-terminus is exposed to the inner nuclear membrane, corroborating the previous findings of the cileviruses MP, to access the cell nucleus [10].

These results, together with the subcellular fractionation assays and prediction analysis, allowed us to propose a topological model of association of the CiLV-C2 p29 and MP with cell membranes. Our model posited that the p29 is peripherally associated with the membrane, while the MP is an integral protein, whereby the full-length molecules for both proteins are oriented towards the cytoplasmic face of the biological membrane (Figure 3viii,xii).

The membrane coupling capacity shown for CiLV-C2 p29, also observed previously for the CiLV-C p29, could justify the capacity of the cilevirus p29 to move along the ER system [7], suggesting their putative involvement in the intracellular viral spread and/or the initiation of viral replication in the newly infected cells [10].

### 3.3. The MPs of CiLV-C2 and CiLV-C Could Have a Similar Membrane Topology

The integral membrane-associated topology of CiLV-C2 MP with both N- and C-termini exposed to cytosolic face suggests the presence of at least two membrane-spanning domains. It is possible that the HR1 and HR3 strongly hydrophobic regions (HR1 ∆G 5.65 and HR3 ∆G 5.33, Figure 1) that overlap in part with the predicted transmembrane (TM) regions could represent the two TM segments, and HR2 could be inserted peripherally to cell membranes. In this sense, it is interesting to mention the presence of charged residues in TM2/HR3, which are rare but can be found in membrane-spanning regions [26] where Lys and Arg, in particular, can be accommodated by their “snorkeling” into the lipid headgroup region [27]. Charged amino acids are also consistently located at the TM flanking regions [26] where basic amino acids act as stronger topological signals than acidic amino acids [28,29], but both types of charged residues are predominantly located near the cytoplasmic end of the TM segments [26], as observed in the proposed model. However, we did not rule out that HR2 could also be a TM segment; thus, we presented all possible models of association of the MP with the membrane (Appendix A). In contrast to CiLV-C2 MP membrane orientation, Leastro et al. [7] hypothesized that CiLV-C MP could be a multi-pass membrane protein with three TM segments, exposing the N- and C- termini to the ER-lumen and the cytosol, respectively. This hypothesis was suggested due to the absence of YFP reconstitution signal from the CiLV-C MP N-terminus to any of the evaluated subcellular compartments, probably due to an incompatible right orientation of the two YFP fragments, and by the prediction of three hydrophobic domains in the protein [7]. Given the cytoplasmic exposure of the N-terminus of CiLV-C2 MP (suggesting the presence of two TM regions) and the partial similarity between both cileviruses MPs (see alignment, Appendix A), it is more likely that CiLV-C MP topology behaves just like the models proposed herein for CiLV-C2 MP.

### 3.4. Cilevirus MPs Are the First Members of the 30K Superfamily Showing a Transmembrane Association Pattern

The CiLV-C2 MP belongs to the 30K superfamily [10]. This family includes MPs from DNA and RNA viruses that show conserved motives, corresponding to seven predicted β-strands connected by putative loops with different sequence patterns and a conserved aspartic acid residue, called the “D motif”, at the end of strand 3 [30]. Although there is similarity between the CiLV-C2 MP topology with other 30K MPs [18,19,21,31,32], the transmembrane association-pattern identified here is rarely noticed for members of this family. Almost all 30K MPs have been presented as proteins peripherally associated with the membrane [18,19,21]; the exception is the genetically related CiLV-C MP [7]. Interestingly, the transmembrane association pattern observed for the MP of both cileviruses has been noticed for viral movement factors not belonging to the 30K superfamily [14,16,20]. The open question is to understand why this model diverges from the rest of the MPs assigned to the 30K superfamily and if this feature results or not in a biological fitness or benefit. It is worth mentioning that, although both CiLV-C and CiLV-C2 MPs have been proven to be functional to rescue the defective alfalfa mosaic virus (AMV), turnip crinkle virus (TCV), and TMV cell-to-cell movement-mutants, including the systemic transport of AMV [10,24], the cileviruses do not move systemically in their natural or experimental plant hosts [4]. It has been hypothesized that members of this genus evolved from an ancestor arthropod virus that became capable of infecting plants after acquiring the movement protein from one or more plant virus(es) [4]. Although we have strong evidence that the cileviruses systemic movement limitation is not due the functional restrictions in their MPs [10], why cileviruses do not have the ability to move systemically within their plant hosts remains unanswered.

## 4. Conclusions

The capsid protein (p29) of CiLV-C2 is peripherally associated with cell membranes with the N- and C-termini exposed to the cytosol. The movement protein (MP) of CiLV-C2 is a transmembrane protein with the N- and C-termini exposed to the cytosol. The amino acid sequence analysis suggested that both CiLV-C2 and CiLV-C MPs could have a similar transmembrane topology. The cilevirus MPs are the first members of the 30K superfamily showing a transmembrane association pattern.

## Figures and Tables

**Figure 1 microorganisms-09-00418-f001:**
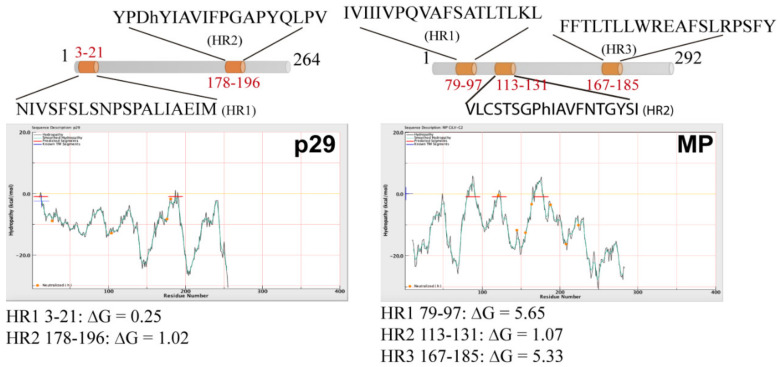
Hydrophobic prediction analyses of the citrus leprosis virus C2 (CiLV-C2) movement protein (MP) and capsid protein (p29). Hydrophobic regions (HR) were predicted for p29 and MP. A schematic representation of the proteins highlighting the HRs can be found at the top of each picture (in orange). Hydrophobic profile of the proteins is shown in the graphics generated with MPEx tool. The red lines show the mean values using a window of 19 residues and the yellow line indicates the predicted HRs. Values of ΔG of the HR are indicated.

**Figure 2 microorganisms-09-00418-f002:**
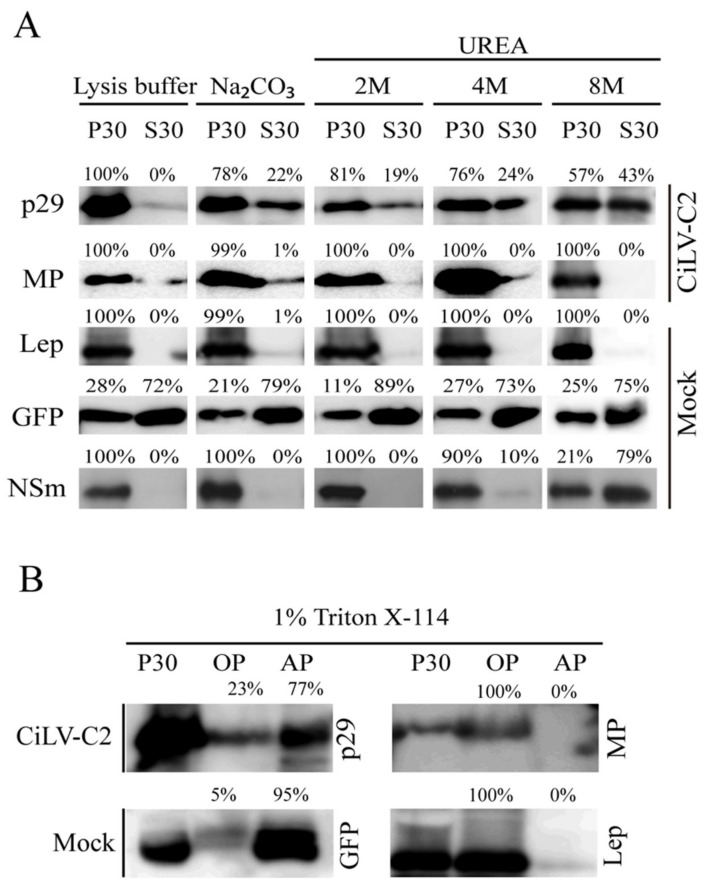
Membrane association analysis of CiLV-C2 p29 and MP proteins. (**A**) Segregation into membranous and soluble fraction of p29 and MP proteins expressed in planta. All analyzed proteins were expressed in *Nicotiana benthamiana* leaves by agroinfiltration. As controls, we used leaf protein extracts containing free enhanced green fluorescent protein (eGFP) (non-membrane), the hemagglutinin (HA)-tagged Lep (leader peptidase) (integral membrane), and HA-tagged NSm of chrysanthemum stem necrosis orthotospovirus (CSNV) (peripheral membrane) proteins. The supernatant from ultracentrifugation after membrane fractioning (S30) and comparable pellet (P30), untreated or submitted to alkaline or urea (2, 4, or 8 M) treatments, were analyzed by western blot using anti-NtGFP antibody or anti-HA antibody (Thermo Fisher Scientific). Relative quantification values are presented. (**B**) Triton X-114 partitioning assay of CiLV-C2 p29 and MP. The P30 fractions subjected to treatment with Triton X-114 were separated in aqueous (AP) and organic (OP) phases. Equivalent amounts of fractions were analyzed by western blot. The GFP and Lep proteins were used as controls.

**Figure 3 microorganisms-09-00418-f003:**
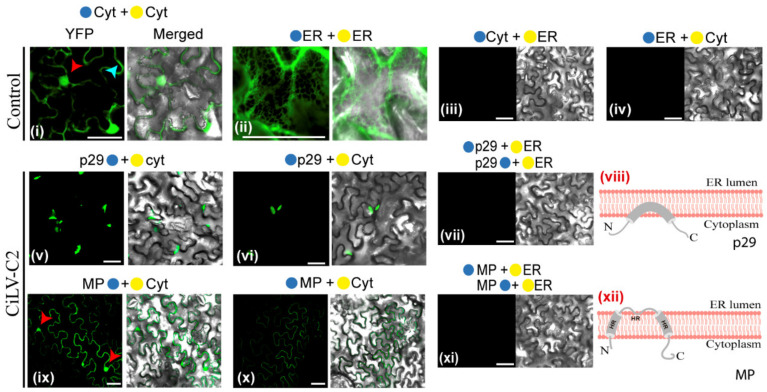
Membrane association and topology of the CiLV-C2 p29 and MP proteins. Subcellular localization (cytosolic face orER lumen) of the N- or C-termini of the CiLV-C2 p29 and MP proteins. The proteins carrying the N-terminal (●) or C-terminal (●) YFP fragments fused at their N-(●/●-ORFs) or C-termini (ORFs-●/●) were transiently co-expressed in *N. benthamiana* leaves with the corresponding complementary yellow fluorescent protein (YFP) fragment addressed to the cytosol face (●-cyt or ●-cyt) or the lumen of the ER (●-ER or ●-ER). Positive and negative controls are presented in the pictures (**i**–**iv**). Images reveal the topology of the C-termini (**v**,**ix**) or N-termini (**vi**,**x**) of the respective CiLV-C2 proteins. Negative YFP signal was observed when the CiLV-C2 p29 and MP carrying the N-YFP fragment at their N- or C-termini were co-expressed with counterpart YFP fragment (C-YFP) addressed to the ER lumen (**vii**,**xi**). Blue and red arrows indicate the cell cytoplasm and nucleus, respectively. Hypothetical topologic models for the p29 and MP proteins are represented (**viii**,**xii**). HR, hydrophobic region. All images contain two pictures corresponding to the YFP signal or merged with bright field. The fluorescence was monitored at four days post-infiltration using a confocal Zeiss LSM 780 model. Bars correspond to 50 μm.

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
