# Peer review of "Membrane Association and Topology of Citrus Leprosis Virus C2 Movement and Capsid Proteins"

_microorganisms, 2021, doi:10.3390/microorganisms9020418_

Round 1

Reviewer 1 Report

In this paper, the authors analyzed p29 (CP) and MP of CiLV-C2 using the methods used in previous papers that have characterized the CiLV-C proteins (Leastro et al. 2019, in fpls). Therefore, the experiments and the results presented here were solid.

However, even after reading the introduction, the significance of using the CiLV-C2 for further analysis of what is already known in CiLV-C is not clear. The significance of this study using CiLV-C2 should be clearly described in the introduction.

Specific comments
1) L49-53:It should be clearly stated that these are the results of the CiLV-C study.

2) It would be better to state the homology between CiLV-C and CiLV-C2 in the introduction . In the original manuscript, I had to read the Ref. 2 to know the difference between CiLV-C and CiLV-C2.

3) L193-194, Fig.2A: In the blots of 8M urea-treated sample, the accumulation rate of p29 in P30 and S30 does not appear to be 31:69 at all. To me it looks like an equal amount of p29 accumulation. How did you measure this?

4) L273, L274: CiLV-C2 CP, CiLV-C CP→CiLV-C2 p29, CiLV-C p29P

5) Fig.S1: Add the TM region of the CiLV-C MP.

Author Response

In this paper, the authors analyzed p29 (CP) and MP of CiLV-C2 using the methods used in previous papers that have characterized the CiLV-C proteins (Leastro et al. 2019, in fpls). Therefore, the experiments and the results presented here were solid.
However, even after reading the introduction, the significance of using the CiLV-C2 for further analysis of what is already known in CiLV-C is not clear. The significance of this study using CiLV-C2 should be clearly described in the introduction.

Answer:
A sentence explaining about the significance of using the CiLV-C2 was added in the introduction section (see lines 65-67). We must emphasize that this type species is currently the most prevalent in Colombia and that this situation could be extended to other South American countries in the near future. Thus, we are convinced that it is important to go further in the knowledge of this new virus species. 

Specific comments
1) L49-53:It should be clearly stated that these are the results of the CiLV-C study.

Answer:
This information was added in the text (see line 49).

2) It would be better to state the homology between CiLV-C and CiLV-C2 in the introduction. In the original manuscript, I had to read the Ref. 2 to know the difference between CiLV-C and CiLV-C2.
Answer:
Thanks for this valuable suggestion. This information was added in the introduction section (lines 54-56).

3) L193-194, Fig.2A: In the blots of 8M urea-treated sample, the accumulation rate of p29 in P30 and S30 does not appear to be 31:69 at all. To me it looks like an equal amount of p29 accumulation. How did you measure this?
Answer:
Thank you for this observation. The values from p29 8M urea treatment were wrong. We reviewed the measurement and obtained new values more suitable. Now the accumulation is presented as 57:43. Percentage values of relative protein accumulation was measured using Fiji ImageJ program Ver 2.0 with ISAC plugin. This information is now presented in the M&M section, lines 135-138. 

4) L273, L274: CiLV-C2 CP, CiLV-C CP→CiLV-C2 p29, CiLV-C p29P
Answer:
It was modified.

5) Fig.S1: Add the TM region of the CiLV-C MP.
Answer:

Computer tools were not consistent to predict transmembrane domain for this protein (see table 1). For this reason, TM are not demonstrated for CiLV-C MP, only hydrophobic regions. In any case, in the revised version of the manuscript we have introduced the predicted TM regions for CiLV-C MP

Reviewer 2 Report

This short paper concerns the analysis of membrane binding properties of two plant cilevirus proteins (coat protein and movement protein). This group of viruses is rather poorly investigated. So the new data on the virus proteins have obvious interest for virologists.

The authors used computer predictions, chemical treatments after membrane fractionation and biomolecular fluorescence complementation to reveal membrane binding ability and topology of the virus-specific proteins. It was shown that both proteins are associated with membranes. Movement protein behaves like integral membrane protein, whereas coat protein is peripherally associated polypeptide.

My main point of critizism concerns author's conclusions concerning topology of movement protein. First of all, I hardly believe that MP region HR3 could be trans-membrane segment. Indeed, it contains three charged residues (two in the central part and Arg in the C-terminal part). So, most probably it can be associated with membrane as peripherical protein piece. Thus, to explain the exposition of the N- and C-termini to the cell cytoplasmic compartment, one can propose two alternative topological alternatives. First, HR1 and HR2 are both transmembrane segments. Second, HR1 and HR2 could be hairpin-like re-entrant membrane-bound segments [see, Betancourt-Solis et al., 2018, J. Biol. Chem. (2018) 293(48) 18514–18524; Kumar et al., 2019, MBoC 30, 1377 (http://www.molbiolcell.org/cgi/doi/10.1091/mbc.E18-11-0698)]. Anyway, the authors should present all possible topological variants for MP with indications of predicted secondary structure of membrane-bound protein segments.  

Author Response

We appreciate the comment and indeed, there is not a definitive topology regarding the different hydrophobic regions determined by the transmembrane prediction software. Our model is based in the two transmembrane regions predicted by MPEx, △G prediction and TMpred programs, which determined that HR3 region is possibly a TM region. The presence of charged residues Glu, Arg, Lys and Asp in TM regions are relatively rare, as stated by the reviewer, but they can be found (Baeza-Delgado et al., 2013; doi:10.1007/s00249-012-0813-9) and Lys and Arg in particular, can be accommodated by their ‘snorkelling’ into the lipid headgroup region (Öjemalm et al., 2016; doi:10.1073/pnas.1606776113). Also, charged amino acids are consistently located at the TM flanking regions (Baeza-Delgado et al., 2013; doi:10.1007/s00249-012-0813-9) where basic amino acids act as stronger topological signals than acidic amino acids (Nilsson and von Heijne 1990. doi: 10.1016/0092-8674(90)90390-z; Saurí et al. 2009; doi:10.1016/j.jmb.2009.01.063) but both types of charged residues are predominantly located near the cytoplasmic end of the TM segments (Baeza-Delgado et al., 2013; doi:10.1007/s00249-012-0813-9), as observed in the proposed model. In any case, we have modified the text to introduce all mentioned comments, specially the presence of basic residues in one of the predicted TM regions (lines 293-302). Also, and as suggested by the reviewer, we indicate all possible models in a supplementary figure, suggesting HR1/HR2, and HR2/HR3 regions as TM segments. Thus, three possible topologies for MP are presented. However, we did not exclude the initial topology presented which suggests that the HR3 region could be a TM segment.